# Port's Role as a Determinant of Cruise Destination Socio-Economic Sustainability

**Maria Santos** [1,*], **Elena Radicchi** [2] **and Patrizia Zagnoli** [2]

[1]   Marketing, Operations and General Management Department, Business School, Lisbon University Institute, 1649-026 Lisbon, Portugal

[2]   School of Economics and Management, Università degli Studi di Firenze, 50121 Firenze, Italy

*   Correspondence: maria.santos@iscte-iul.pt

**Abstract:** This article argues that the cruise terminal ports play a crucial role in the economic and socio-cultural sustainability of destinations, bridging the onshore tourism offered among cruise companies, global operators, and local business and infrastructures. They support the promotion of local brands and reduce congestion. The impact of crowds on the identity of coastal cities triggered the attention of academia and media, alerting for their negative impact, specifically from the Mediterranean cruises. In parallel, it raised the research interest on cruise tourism carrying capacity and ports planning the integration of cruise tourists' flow. However, previous studies focused on the residents' and passengers' perception of a specific destination, neglecting the port management role. This study aims to clarify the underneath dynamics that allow sustainable cruise–land visit. Employing a qualitative case study approach, it compares data obtained from secondary sources and port executives' structured deep interviews from two leading transit ports connected with the Mediterranean. Lisbon is amongst the most popular tourism destinations and international cruise terminals; Livorno is a gateway port to Tuscany, mainly Florence and Pisa. Despite their different patterns, in both ports of call, a strong concern with sustainability and a reduced congestion effect are observed from the management actions on promoting the local offer and on revitalizing the terminal infrastructures in order to provide comfort shopping and entertainment amenities to passengers.

**Keywords:** sustainability; responsible tourism; transit port; port of call; Mediterranean cruise destinations

## 1. Introduction

There is a rising trend toward larger and more frequent cruise ships, showing the globalizing nature of the cruise industry. The literature on cruise tourism highlights the perception of a cruise ship as a "floating hotel" [1], as well as an example of a "tourist bubble" [2], where cruisers enjoy tranquility and staying in a safe environment. Each ship hosts passengers of diverse nationalities with a wide range of services including hospitality, entertainment, and itineraries, which take place both onboard (concerts, theatres, activities, shopping, etc.) and inland (excursions, shopping, visiting historic sites, museums, etc.), under a controlled, safe, and pleasant environment. Nevertheless, passengers desire to discover different opportunities, which may be offered also at the destination ports and not only on board [3]. The cruise companies build their competitive offer by putting together passengers' tourist needs and the appeal of ports' brands, attractions, and inland assets, through relationships developed with several stakeholders (port authority, logistic companies, tour operators, etc.).

The increasingly frequent flows of cruise excursionists affect the ports of call significantly, where ships are usually docked less than one day, requiring port authorities and port cruise terminal managers to engage in active outreach, not only with the cruise companies, but also with local

businesses. In particular, in mature and/or historic destinations, the services and facilities of the ports are the cruisers' first contact with the city, as well as the first protection and gateway from the cruise ships to the city. Therefore, they are an important stakeholder given the complexity of achieving a pleasant and sustainable experience, while keeping the balance between the interests of the residents, the visitors, and the cruise ship industry [4].

The research on cost/benefit trade-off is scarce, as the studies addressing environmental sustainability are still limited and far from being well established [5]. The debate on the major positive and negative consequences of cruise tourism is still not consensual in most venues of impact analysis, namely, environmental, socio-cultural, and economic. Both research and media show contrasting perspectives regarding the socio-economic benefits of cruises in general, whereby the benefits of cruise tourism are geographically concentrated in locations attracting excursions and tourist walks. Seminal studies emphasized the positive impacts while acknowledging the environmental costs and large asymmetries between the local benefits and national spillovers [6]. Recent case studies such as Nanaimo in Canada [7] and Napoli in Italy [8] illustrate the importance of a "responsible cruise tourism" vision leading the port governance to handle the interrelationship with the community stakeholders in order to capture socio-economic benefits. In general, the literature shows that coastal residents and local businesses accept the coming of huge cruises ships as they bring economic development, although some studies also identified that, for the residents, the high expectations of potential—and sometimes promised—benefits from the cruise port were not met [9].

The cruise tourism research remains quite fragmented and based on economic impact studies prepared for industry stakeholders [10]. Regarding destination planning and attraction, cruise tourism research analyzed the residents' perception [4,9,11–14], the tourists and cruise passengers' crowd perception and satisfaction [15–17], the destination communities' driver and stakeholder interrelationships [11,18,19], or the ports' strategies of carrying capacity and competitive factors [7,8,20,21]. Studies highlighted the importance of ports developing marketing strategies to promote lengths of stay [22], the port terminals' factors that matter to the cruise ships and passenger itinerary choice, such as the infrastructure [23], the integration of the port and city [8], the services offered [24–26], the experience of the "local flavor" [27] or the cultural capital [11], the local business socio-economic value [9,20,28], and the revitalization of the ports, in order to increase the comfort of the embarking/disembarking which may be difficult to access and, thus, an unfriendly place, eliminating port choice [29].

Many port cities in the Mediterranean enjoy the benefits of cruise tourism, triggering their strategic positioning as "tourist ports" [24], leading some, like Lisbon (Portugal) and Livorno (Italy), to invest significant amounts in building or rebuilding cruise terminals [7,8,25]. In spite of the positive assessment by industry associations and the European Commission reports, there is concern about the loss of historic towns' authenticity and the congestion, especially in the most crowded towns targeted by different transport alternatives. We think of Barcelona, Venice, and Lisbon, but this even includes Florence, which is reachable via cruise dock in Livorno.

The present article attempts to enrich the understanding of the port managers' contribution to destination competitiveness and the asymmetric framing of cruise stakeholders, i.e., the active participation of the port authorities and the cruise terminal managers in pulling communications of local businesses and terminal facilities. Given that the influence of the ports (and port cities) and the port terminals on the cruise organization's offer is understudied [30], this study innovates because it includes the port managers' perspective, as well as observing the port managers' actions in order to integrate local businesses in the terminal cruise facilities and in the onshore tour range. Although local businesses may contribute to the destination competitiveness in parallel with the proximity of world touristic attractions [30], excursions are mostly bought on board via global travel operators, which the port may have difficulty in controlling. This study contributes to the analysis of how port managers may bridge the relationships between local businesses and global companies, as well as attract land visitors to terminal cruise facilities, thereby helping to reduce tourism congestion.

A comparative–qualitative approach is followed, supported by secondary data, observation of the ports' infrastructures, and deep interviews with executives of tourism public entities and the ports' administrations. The comparison is between two ports of important historical areas (Lisbon in Portugal and Livorno within the Tuscan region in Italy), located at the beginning and in the middle of Mediterranean tours, both facing rising mass tourism, which provides a rich context for the study of this phenomenon. Some differences are evident. Lisbon is an important destination town, the capital of Portugal, where cruise tourism awareness is quite recent, but consciously managed by the local municipality in collaboration with a variety of stakeholders (port authority, international associations, local businesses, etc.). Livorno is a "middle" town, traditionally a trading and ferry harbor which developed into the role of "gateway" toward inland attractions, such as Florence and Pisa. In common, both ports are investing in infrastructure innovation and in cooperative relationships with industry associations, cruise companies, and local businesses in order to reduce the tourism flow and the local consequences of the oligopolistic role of global cruise companies and travel business agents.

This introduction presents an overview of the literature reflecting the trend of the environmental, economic, and socio-cultural sustainability of ports of call that are in Mediterranean coastal historic towns. Section 2 provides a literature review examining the complexity of stakeholders in cruise itineraries, the port typology, and their impact offshore, onshore, and inland. Section 3 proposes a framework analysis reflecting the factors that may be considered in port management in order to reduce the destinations' dependence on the cruise industry's global players, necessitating a strategic collaboration among different actors in order to gain an advantage for every player involved: the port, the cruise line, and the destination. In Section 4, the materials and the qualitative case study method are described. Finally, after presenting the results, the discussion and concluding remarks are drawn, which appoint the research limitations and managerial and academic implications.

## 2. Literature Review

### 2.1. The Development of Cruise Tourism

Cruise tourism is very peculiar since it combines "tourism" and "hospitality" to offer a whole leisure experience while traveling from different coastal towns [31,32]. Today, the segments and product ranges in cruise ships are quite large and diverse, a far cry from the cruise concept as a holiday targeted to seniors searching for a relaxed experience, and a social and cultural combination. Contemporary cruises are affordable for families and even for millennials traveling alone.

Over the 10-year period, from 2007 to 2017, global tourist arrivals, mainly land-based tourists, rose from 930 million in 2007 to an estimated 1323 million in 2017, approximately 42% overall [33]. The international demand for cruises had an annual growth rate of 5.4% corresponding to an overall rate of 68.5%. European cruise tourism increased by 71.9% versus the North American increase by 25.6% [34]. After the Caribbean, Europe is the world's second largest cruise ship destination. Port-of-call passenger visits rose by 22% over the 2009–2014 period, growing from 23.76 million to 28.96 million [35].

According to the Med Cruises report [36], a substantial growth in passenger movement happened over the last decade: the 10-year growth stands at 27.6%. At the beginning of the century, this number stood at 4.3 million passenger movements, confirming the growth that cruise activities in the Mediterranean experienced since then. The ratio of transit passengers to passengers' homeporting from Med Cruise member ports stands at 71/29. This ratio remained stable over time, as it was almost the same throughout the last 10 years.

Mediterranean destinations led the mentioned worldwide growth, recording extraordinary results with 8% more international arrivals than in 2016. As a result, the Mediterranean stands today as the second biggest cruising region, following the Caribbean. Combined, the two major cruise regions, Caribbean and the Mediterranean, host 51.2% of the global cruise fleet capacity. In the Mediterranean, the scale of cruise passengers exceeded 27 million movements per year three times during the last five years, with 2013 being the first time ever that this total exceeded 27 million movements [36].

## 2.2. Coastal Destination Cost/Benefit Trade-Off

The fast increase in the number and diversity of passengers, in parallel with the bigger dimension of ships and the large range of their entertainment offers, drew attention to the need for a congestion management approach, not only with regard to the host destination, with port and cruise industry stakeholder analysis [7,8,19,21,28] and the perceptions of the residents and the local businesses [11,12,14], but even with regard to the cruise passengers [15,17].

Cruise tourism is just one of the several coastal tourism segments, but it is the fastest growing and the most dynamic, showing an increasing trend and resilience to the economic crisis [37]. According to this study, the global economic impact of the cruise industry is high in Europe and especially in the Mediterranean regions, where Italy holds a consolidated leadership position. Italy is the largest cruise ship manufacturer in Europe and the largest turnaround port country.

Over the past 10 years, the cruise industry expanded over 69%, exceeding land-based tourism (42%) [34]. The cruise industry generated about 137 million passenger and crew visits at ports around the globe. European ports count for less than half of the number of passengers and crew from the North American leading destination. By purchasing pre- and post-cruise vacations, shore excursions, souvenirs, and other retail goods, passengers spent an estimated $17.7 billion, representing 29% of total cruise sector direct expenditures. In 2015, the cruise industry generated about 16.9 billion euros derived from four sources: the passengers, the ships' procurement of goods and services to support their operations, the compensation of the administrative staff of the companies and crew, and finally the construction and maintenance of cruise ships, which alone accounts for around 15% of the total direct economic contribution.

The socio-cultural impact and the risk of cruise congestion are being discussed in academic research in spite of the increase in the global economic contribution of the cruise sector. According to the Cruise Lines International Association (CLIA) (2018) [34], the combined direct, indirect, and induced contributions generated by cruise tourism were estimated to be $134 billion in 2017, showing an increase of 6.3% from 2016. Although the European Commission recognized the economic impact of cruise line activity and its contribution to the European Union (EU) economy [35], many researchers questioned the short-term economic benefit to the destination [19]. Johnson (2002) [38] estimated that cruise passengers' outlays in local economies are small. Studies showed that the economic benefits of cruise tourism are greater in homeports or turnaround destinations [39]. Others revealed that, despite the existence of fees (docking and passengers) and revenue from visitor and crew expenditure on souvenirs, food, and shore excursions, the economic benefits are typically less than for land-based tourism, since stopover tourists spend on average 10–17 times more than cruise ship tourists [11,28,40]. Local benefits failed to materialize when cruise tourism was undertaken without investment in an involvement of destination communities, when comparing four towns close to a new port in Trujillo, Honduras [11].

Regarding previous studies [11,39,41], economic benefits are not so evident as the cruise tourists tend to eat, sleep, and even book onshore excursions on board, as in some coastal towns the tours are organized to other nearby towns which are more historic or interesting. Cruise tourism may be an important development driver for port-cities, depending both on the operational profile of the market and on the domestic conditions, such as the size and facilities of the ports [19]. Particularly for places in a low economic context, the socio-cultural impact should also be a key factor to consider before proceeding with big investment for cruise terminals or increasing the wharf dimensions to receive bigger vessels [21]. Nevertheless, accurate studies on the local impacts of cruise tourism are still rare [8], and the studies measuring the cost/benefit trade-off are still in their infancy [41].

## 2.3. The Cruise Sector Structure and Drivers

The contemporary cruise industry began in the late 1960s and early 1970s with the founding of the Norwegian Cruise Line (1966), Royal Caribbean International (1968), and Carnival Cruise Line (1972), which emerged as the largest cruise lines. The early goal of the cruise industry was to develop

a mass market since cruising was, until then, an "elite" activity. A way to achieve this was through economies of scale as larger ships were able to accommodate more customers, as well as creating additional opportunities for onboard sources of revenue [34].

By the 1980s, economies of scale were further expanded with cruise ships that could carry more than 2000 passengers. The current large cruise ships have a capacity of about 6000 passengers, but the bulk of cruise ships are within the 3000–4000 passenger range. The market for the cruise industry was by then established and recognized as a full-fledged touristic alternative directly competing with well-known resort areas such as Las Vegas or Orlando.

The market drivers of the contemporary cruise industry are similar to those that fostered the growth of tourism after World War II, particularly the rising affluence of the global population and the growing popularity of exotic and resort destinations. For some analysts, what is novel with cruising is that the ship represents in itself the destination [42], acting as a floating hotel (or a theme park) with all the related facilities (bars, restaurants, theaters, casinos, swimming pools, etc.). This permitted cruise lines to develop a captive market within their ships, as well as for shore-based activities (e.g., excursions or facilities entirely owned by subsidiaries of the cruise line).

As described by Rodrigue and Notteboom (2013) [43], the cruise industry has a very high level of ownership concentration, since the four largest cruise shipping companies account for 96% of the market as measured by the number of passengers (Carnival Lines, Royal Caribbean, Norwegian Cruise Line, and Mediterranean Shipping Company—MSC Cruises). High levels of horizontal integration are also observed, since most cruise companies acquired parent companies but kept their individual names for the purpose of product differentiation. For instance, Royal Caribbean Cruises, the world's second largest cruise company behind Carnival Lines, accounts for 24% of the global market serviced under six different brands such as Celebrity Cruises (which caters to higher-end customers) and Azamara Club Cruises (smaller ships servicing more exotic destinations with shore stay options) [34]. The cruise industry, thus, presents an illusion of diversity with the bulk of the market firmly in the hands of large players.

The cruise industry over time became oligopolistic as high levels of concentration emerged [44]. Although the penetration into new markets occurs through alliances and collaboration with local brands, there is a dominant power of the cruise companies and a concentration of itineraries, leading to the overload of a small number of ports, whether homeports or ports of call [19].

The organization of onshore itineraries is difficult for the host ports to control due to the increasing asymmetry of bargaining power between port managers and cruise ship operators. The importance of a port can, therefore, be different based upon the commercial strategies of its users, primarily, in this specific point of view, the cruise companies.

The cruise industry sells itineraries, not destinations [45], underlining the core importance in the selection of a sequence of ports of call. Cruise operators are challenged to develop competitive cruise packages but, at the same time, they must optimize the deployment of their cruise ship fleet in view of minimizing operating costs and/or maximizing revenue per passenger slot. As such, vessel deployment strategies and itinerary design are affected by market circumstances and requirements such as the seasonality in demand [45], the optimal duration of a cruise vacation, the balance between sailing time and shore time, the existence of "must see" destinations, and overall guest satisfaction. Cruise lines adapt itineraries to different regions and passenger segments [42]. The itinerary and the port infrastructures are key factors for the cruise owner's decision [23]. Also, the length of stay of cruise ships in ports is influenced by the attractiveness of the port of call and the distance between the previous and the following ports [22].

Several port-cities are investing in order to fit the tangible basic requirements imposed by the cruise companies. Competition among coastal cities to be part of the cruise market is fierce; thus, a region/destination/port needs attractive, special, unique, or iconic characteristics to attract cruise lines and get cruise passengers from abroad. There are, therefore, important intangible requirements such as the destination's brand, reputation, and inland potential tourism attractiveness, where port-cities

and local institutions should invest if they would like to be part of the global cruise circuit. Observing that trend, Rodrigue & Notteboom (2013) [43] suggested that a next step will involve the development of new cruise terminals, and a closer integration between the cruise port and the cruise line.

Being part of the "global cruise circus" is not always a "must". Each port-city should base its own market position according to several variables, usually linked with the history, cultural background, socio-economic humus, social milieu, strategic vision, mission of the local government, and so on.

Therefore, several governments and port authorities, especially those in developing countries aiming to become new tourism destinations, are investing massively in re-qualifying, building, or extending cruise terminals [8,11], even if entry barriers are very high.

### 2.4. Port Categories and Their Role in the Cruise Destination Developement

Ports have to meet some key requirements of cruise lines in order to be considered potential cruise destinations (ports of call). Indeed, the specific cruise sector structure implies that not every port can be suitable and included in this particular segment of tourism. To be considered a port of call, there are some tangible requirements [46], such as the existence of a cruise terminal or an alternative docking facility, docks of sufficient length, water of sufficient depth (cruise ships generally require between 8 and 9 m of water to operate safely), the possibility for cruise ships to turn around, a constant level of access regardless sea conditions, good facilities at the terminal or docking facility such as luggage handling space, gangways, parking area, airlift, customs area, waiting facilities, toilets, and information centers, and professional, qualified ground handlers such as inbound tour operators and transport operators. Capacity building is, therefore, very important for destinations that consider developing cruise tourism, involving at least an international airport in the region where cruise passengers can be flown in and out (in the case of "fly and cruise"). Competitive pricing is another important issue, as cruise lines focus on the balance per port when developing itineraries, taking into consideration excursion revenues, port fees, tugboat tariffs, taxes, and agency fees. Safety and security requirements represent another factor, as a port must be able to accommodate cruise ships and their passengers safely. Because of its international nature, the cruise tourism industry is subject to the mandates and guidance of the International Maritime Organization (IMO) which is responsible for establishing international standards for cruise ship safety, design, and construction.

The actual categorization of ports at the international level defines the port's role according to specific dimensions and dynamics between industry stakeholders, such as the cruise companies, the ports, the passengers, the international trade channels, and the global travel agents. The competitiveness of homeports in the Mediterranean actually show different strategic priorities in investment and marketing [20]. London & Lohman (2014) [18] provided a theoretical contribution of the relationships between the cruise destination stakeholders and the cruise companies. The authors identified key stakeholders and the power that underpins their commercial relationship in the context of the cruise industry. Their proposed framework identified five elements that guide the destination's development: (1) the type of port (homeport, port of call, or hybrid); (2) the stakeholders and their interest (cruise line owners and operators, gatekeepers as regulatory officials and transport providers, the port-side stakeholders as the port owners and operators and ship service providers, and the shore-side stakeholders as the government, investors, tour operators, and local transport and other business providers); (3) the stage of development of the cruise destination; (4) the port characteristics; (5) the origin of the proposal for cruise infrastructure.

Marti (1990) [47] classified three port categories according to their position in the cruise itinerary, for which the investment required is different: the homeport (or turnaround), the port of call (or transit port), and the hybrid port.

A port of call is an intermediate port where ships customarily stop for supplies, repairs, or transshipments of cargo. As it relates to the cruise industry, a port of call is a stopover destination included in an itinerary.

A homeport (or turnaround) is the starting and/or ending point for a cruise itinerary. There are some major conditions that a cruise port must fulfil in order to become a home port. The first condition is the presence of adequate port infrastructure (operational depth at the dock, the length of the pier, the existence of a passenger terminal, etc.). The second one is the efficient provision of an extensive range of services to the cruise ship, the passengers, and the crew: security equipment, warehouse and baggage handling equipment, parking area for coaches, taxis, and private autos, supply provision, and ship repairs. The third condition is the connectivity with other transport modes, such as the existence of a well-connected international airport, the existence of a train station, and the connection of the cruise port with road networks. The fourth condition is the ability of the port-city to host the cruise passengers. Most cruise passengers choose to stay at the port-city prior to their embarkation or after their disembarkation from a cruise ship. As such the port-city must have the necessary infrastructures able to accommodate the cruise passengers. These infrastructures include hotels and restaurants. The hybrid port respects both sets of characteristics. The homeport is also referred to as a hub port [45], although in the sense that its demand is very high.

A hub port is a central location in a transportation system with many inbound/outbound connections of the same mode. The hub-and-spoke system is growing as a result of the advent of large vessels in the cruise industry. Whereas, in the past, most vessels stopped over in all route ports, large vessels are only stopping at large hub ports where anchoring would be feasible. This translates into an increase in the so-called transshipment freights. Generally, in logistics, the freights are unloaded in the main hub in the territory, and then the small vessels carry the freights from the hub ports to the neighboring ports.

The ports may be classified into three categories depending on the role they serve within their regions: destination cruise port, gateway cruise port, and balanced cruise port [43].

A destination port is a place where the city overlaps with the tourist offers. It is usually a "must see" city that cruise companies wish to include in their itineraries. There are several reasons why the cruise port area can be the sole destination. In the case of cities such as Venice and Barcelona, the cultural amenities offered are world class to the point that tourists have little incentive to see anything else in the vicinity. The cruise terminal and its immediate area essentially act as a tourist bubble [2].

A gateway port is a location (terminal) where major flows of passengers, goods, and ships converge. It is a transit place, where carriers stop over for oil and other service procurement. It has many inbound/outbound connections of different modes (e.g., maritime and land). At the same time, the tourist destination is not the port city, but the surrounding territory. It is, therefore, a kind of "corridor" to reach other inland attractions.

A balanced port is a location quite attractive, where the port can be a destination, but where excursions to other places not far are also available.

## 3. Materials and Methods

### 3.1. The Proposed Framework

The literature shows that many ports are undertaking investments in their berths, maritime stations, reception logistics, transportation, events, inland connections, and so on. The focus of our research is the influence of the port authority and cruise terminal management on the sustainability of cruise service management when the boats are docked. Sustainability should be achieved through investment and socio-economic decisions that preserve local identity from standardization and congestion.

For sure, the services and the infrastructures necessary for a port of call are different from those of a homeport, a destination, or a gateway. The cruise companies choose the port to touch and dock according to logistics and other factors [18]. The competitive dynamics in the geographical area and the target customer see the cruise company building the cruise service from one side to pick up the opportunities of the chosen port's infrastructures, accessibility, positioning, facilities, cruise tariffs to dock, reputation, brand promotion, and popularity of the port's name. However, ports that decide to

interact with and play a role in cruise tourism organize their assets and investments in order to match the cruise companies and the passengers' demand.

Since the cruise sector development is recent, different ports show a clear imitation process. Moreover, according to the scope of cruise tourism, specific stakeholders in the territory are involved to enrich the supply with local identity content and experiences able to overcome the flat mass tourism flows. The trend is to involve cruise industry stakeholders as terminal operators [43]. The development of associations such as CLIA or Med Cruise allow for much more dynamism, as well as the trade international fairs, where port managers can present their natural, cultural, and technical characteristics, as well as promote local businesses and receive financial and other support. It is important that the attractive capability of each port is not passive and dependent upon the oligopolistic power of cruise companies. Ports must be able to identify their specific and unique advantages to bargain with travel global agents and cruise companies. The port cruise terminal business model is based on this bargaining power, its association membership, and the offer of different services, reflecting its uniqueness, either through the shopping services, or through the entertainment or cultural characteristics. Thus, port investment policies should take into consideration the interactions portrayed in Figure 1.

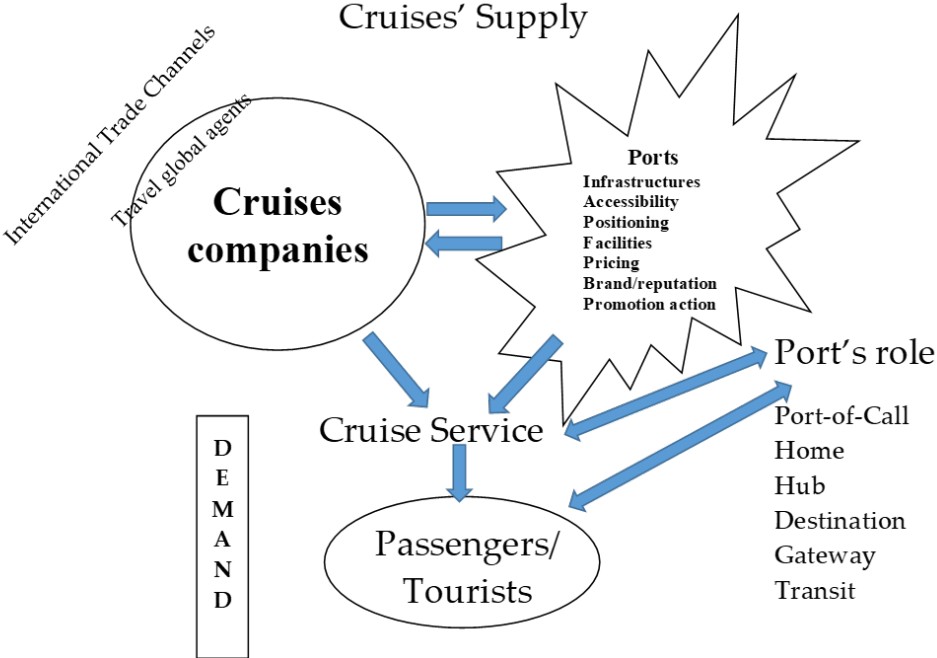

**Figure 1.** Cruise supply strategic process.

Moreover, each port of call needs to be aware of the fierce competition with other nearby ports and with the other ports of call that compose the cruise itinerary, as well as their co-location. Port characteristics and their services offered have a strategic role in ensuring the inclusion of peripheral ports within the cruise network. If a port does not have attractive natural/historical characteristics or a wide range of services, it may nevertheless become a cruise destination, due to its proximity to world touristic attractions and the organization of excursions [30].

The ports we chose to analyze—Lisbon and Livorno—have different specific roles. In spite of its ocean location, Lisbon is a key cruise destination, connecting the Mediterranean and the Atlantic coast. Livorno is a key stop—a "gateway", as the Carnival cruise company calls it—close to very attractive touristic destinations like Florence and Pisa, and it is one of the main attractions in cruise packages in the central Mediterranean Sea.

According to the literature review and the abovementioned meta-analysis, we developed a tentative theoretical framework for analyzing the role of the ports for a sustainable supply (Figure 2). As members of the industry, according to the itineraries and their strategies of development, port

managers act as ambassadors of the town and region, as well as promote local brands and cultural tours. They may reduce congestion by interfering with ship schedules or length of stay, acting as an interface between global tourism operators and cruise companies with local businesses.

The framework incorporates the technical, logistic, and economic features of the port, looking at the socio-economic dimensions and the relationship with the stakeholders that promote the destinations and the inland experience. Indeed, cruise companies will choose destinations according to logistic and other factors [18]. The cruise terminal ports' governance influences the cruise companies' choice, depending on their infrastructure characteristics and the promotional campaigns regarding the destination sightseeing uniqueness [7,8,19,21,48].

## Ports

- Port location (geographical/town)
- Ownership (private vs public)
- Port's core business (trade, raw materials, energy products, ferries, cruises, etc.)
- Scale and Scope Economies;
- Cruise terminal positioning;
- Regulatory control of passengers;
- Port charges and discount: passenger taxes for longer stays and vessel taxes for operators
- Ports infrastructure/innovation/design
- Terminal features: general shops, local products retailers, bar/restaurants, wi-fi, air conditioning, bureau exchange, etc.
- Terminal Logistics: shuttle bus, train, taxi, airport connection, parking, car rental;
- In land assets (proximity to historic town, shopping centers, museums, archeological sites, 'zero-mile' products);
- Additional services: hotels, meeting centers, wellness centers, travel agents, etc.

## Local stakeholders

Port's Authority
Local Government
Tour Operators
Travel Agents
Local businesses

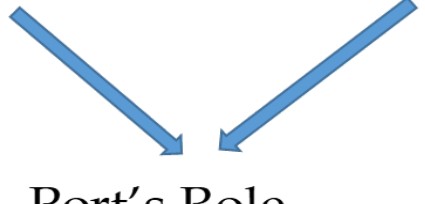

## Port's Role

**Figure 2.** The theoretical framework.

The ports' bargaining power depends on the industry associations that they can gather to support the investment. Both these associations and the ports' governance will take part in the commercial and sales fairs of the global industry, to promote the destination ports and the local businesses. Cruise companies make direct contact with these marketing strategies and may achieve complementary partnerships with the local businesses. The ports may not directly sell onshore excursions, but they should actively promote the competence and differentiation of the local businesses.

### 3.2. Research Methodology: A Qualitative Case Study Analysis

The understanding of the abovementioned relationships was achieved through an exploratory qualitative method based on a comparative case study, the data for which were collected from structured deep interviews addressed to port managers, as well as observations at the cruise terminal, combined with studies gathered by port offices and the public published by consultants, the media, and other secondary data. Two mainly transit ports were compared based on their offer of tourism services: Livorno in Italy and Lisbon in Portugal.

To build the Livorno case study, the main secondary dataset used was the Instituto Regionale Programmazione Economica della Toscana (IRPET) report "Cruising in Livorno and its economic impact on Tuscany" [37]. The IRPET report includes the port operation statistics, services offered to tourists, the integration and accessibility of the territory, and a benchmark analysis from the main cruise ports, as well as the results from a survey addressed to 2288 passengers onshore, chosen randomly on the basis of a sample pre-stratified by the tourists' nationality, rating of the ship, and time of year. This sample covered 77% of the passengers who went ashore from the 807,935 passengers of 403 ships that were docked in the port of Livorno in 2016.

The Lisbon case study was mainly based on the cruise passengers' survey coordinated by Tourism of Portugal and the port report available on its website [49]. Also, industry reports were used, such as the one produced by Deloite [50]. Lisbon data were obtained from a survey conducted on 998 passengers from 52 ships docked in Lisbon from April to November 2016 and 1003 passengers from 49 ships for the corresponding period in 2017.

### 3.2.1. Lisbon: A "Destination" Port

The Port of Lisbon (Porto de Lisboa) is a wide European port and the largest in Portugal, located at the interface between the Atlantic Ocean and the vast estuary of the Tagus, 362 nautical miles away from the Gibraltar strait [49]. where the Mediterranean Sea begins. Its geo-strategic centrality and a water basin of 32,000 ha, sheltered and deep, give the Port of Lisbon a high stature in the logistics chain of international commerce and on the main cruise circuits, offering the best navigating conditions both for large ships of great depth, namely, transoceanic vessels, and for nautical sport. Integrated in the trans-European network of transports, it is the "meeting port" of maritime, railway, and road transport. The Port of Lisbon is still mainly a port of trading general cargo (56%) and mineral and food raw materials (15%), but the passenger traffic is growing, accounting for 15% of the port's business [49].

Together with its own favorable geographical position and a population of around 510,000 inhabitants, the metropolitan area accounts over 2.8 million people. Lisbon is an important town to visit and a destination port with a cruise harbor layout able to easily host the cruise ships. The city serves as a cultural hinge with the Atlantic coast of Europe, the western Mediterranean, and northern Europe, as well as Africa, Madeira, and the Canary Islands. Previous studies focusing on the cruise passengers' motivations in Lisbon found that they considered it a cultural visit [51].

In the last decade, tourism (and cruise tourism) expanded significantly. The town became a very successful touristic destination for visitors, who arrive mainly by plane (94%) from all over the world [50]. The number of foreigners staying in hotels in the Lisbon metropolitan area was more than four million, where 70% stayed in the city.

Lisbon has three cruise terminals, which easily allow direct immersion into the town. At the *Santa Apolonia* terminal, opened in November 2014, and *Jardim do Tabaco Quay*, cruise passengers can walk directly to the center of the city.

Close to the cruise terminal, there are four recreational docks—the Alcantara Dock, the Belem Dock, the *Bom Sucesso* Dock, and the *Santo Amaro* Dock—with security equipment (hand and cabin luggage X-ray machines), able to offer a wide range of services such as tourist information, a post office, public phones, public toilets, souvenir shops, wine shops, car park, coach park, taxis, and a shuttle to the city center. Lisbon supplies approximately 1500 m of quay with depths between 8 m and 12 m allowing the berth of cruise vessels from the smallest to the largest. In order to increase the capacity, the port authority is undergoing works to add 670 m of berthing quay near *Santa Apolónia*.

Figure 3 shows that Lisbon differs from Livorno as a gateway port. For passengers, the city of Lisbon is the prime destination and they rarely undertake further excursion. A large majority of cruise tourism reaches Lisbon as an elective destination where visitors can easily walk downtown. Considering the short amount of time of cruise stops, most tourists prefer to visit the city, on their own or through a guided tour, to discover the town in an organized way. Regarding the passengers who responded to the survey in 2017 and 2018, only 26% bought a guided tour before landing and fewer

than 30% chose excursions out of Lisbon city. In this case, the most visited places were Cascais (18% of respondents) and Sintra (28% of respondents), in the metropolitan regions of Lisbon and Obidos (21%) and Fatima (10%), where the latter is a religious destination.

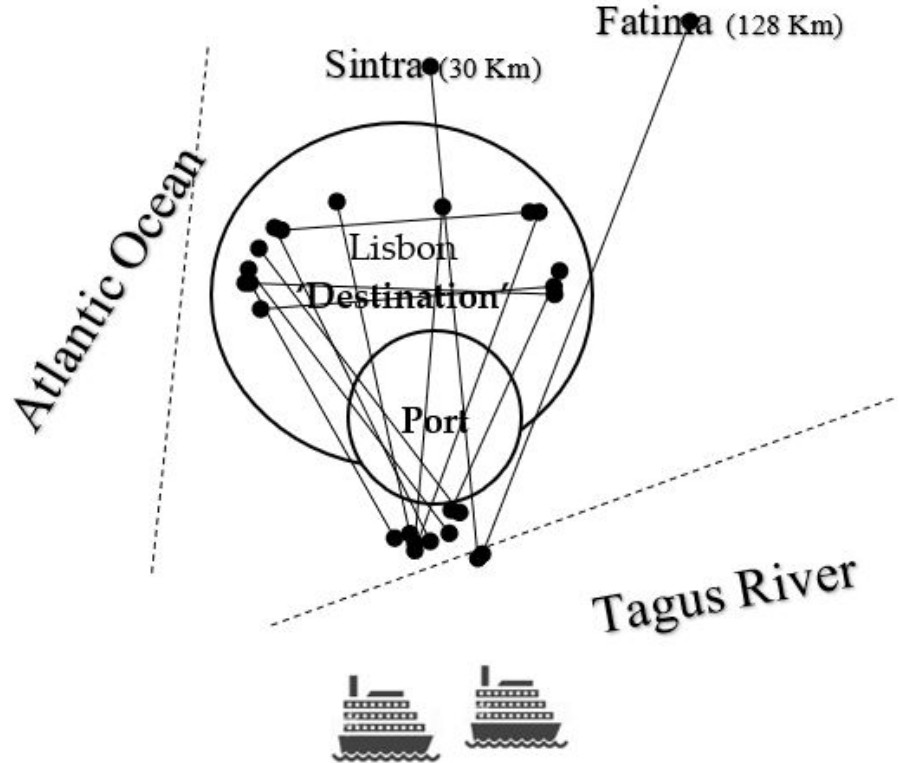

**Figure 3.** Lisbon as a destination port.

Although Lisbon hosted cruise ships for a long time, only in the last decade did the municipality and port authorities consider the cruise business as strategic. Since 2014, the concession of cruise terminal management was granted to a private association, pursuing the so-called "landlord" model: the port's jurisdiction is managed in order to open the port's area to the city. Management, coordination, facilitation, and essential promotions for the maintenance and improvement of the competitive levels of the port and the partnerships are entrusted to the Lisbon Port Community. Both banks of the Tagus gave rise to many restaurants, bars, and outdoor cafés. An important part of the Lisbon nightlife also takes place by the river. Today, the merchant port of Lisbon has a large area that is a stage of entertainment and culture, hosting musical concerts, with both open air and covered spaces.

From January to October 2018, the Port of Lisbon hosted 281 cruise ships and about 487,000 cruise passengers, rises of 1% and 11% compared to the previous year. Between October 2017 and 2018, the number of cruise passengers in transit grew by 48%, while the turnaround went up by 43%.

The strategy of the "landlord" model is to make the Port of Lisbon [49] the following: (i) a functionally diversified port, with three core activities—container cargo, bulk agri-food stuffs, and tourism and leisure—closely tied to the development of the Lisbon metropolitan region and the surrounding area, which will form a potential hinterland; (ii) an integrated port, in harmony with the surrounding areas and city life; (iii) a comfortable and easy destination. In the frame of this strategy, the Port of Lisbon is a member of global associations, such as CLIA, Cruise Europe, and Med cruises, where international stakeholder relationships are considered by private companies that manage the port and by the port authority. The Port of Lisbon negotiated an extension of the number of days the cruises are docked. The extension of the stopover in town is an effective strategy aimed at reducing congestion carried by short-term tourism. The local government, together with national and regional

stakeholders, is attempting to activate a shared policy in order to respond to tourism crowding and its effects on the local communities and environment.

### 3.2.2. Livorno: A "Gateway" Port

The town of Livorno compared to other millenarian areas in Tuscany is relatively "new", as it was started as a port to serve Pisa. The historical maritime republic located at the outfall of the Arno River slowly lost its sea front. Consequently, the coast where Livorno was built in the 14th century became the port of Pisa. The importance of this port for the entire Tuscan region introduced the Lorena Duke in 1575, entrusting Bernardo Buontalenti to design the port enlargement and the new town of Livorno.

At the beginning of the 1600s, by promoting a special hosting legislation called "*Livornine*" that encouraged free cult and free commerce, many Jewish, Portuguese, Greek, Dutch, etc. travelers started moving in from the Mediterranean and European countries. Livorno in a few years became quite a significant multiethnic port town with a key role in serving Florence, the Tuscan region, and central Italy. Venetian technicians able to remove land water and build huge dry dock storage supported the town and its port's growth.

At the end of the 1700s, the Duke of Tuscany liberalized maritime trade and Livorno became an important "free port" with 30,000 residents, where each community could practice their cult, tradition, and cultural activities. It became very significant for international trading between Holland, Europe, and the Mediterranean countries.

The actual town of Livorno accounts for 135,000 residents and it is the second town of Tuscany. This short historical reference points out the following:

- Livorno was born as a trading port supporting all the inland Tuscan economy;
- Pisa at the beginning and obviously Florence were strategically the main reference points for political and economic decisions;
- Livorno is a strategic location for connecting the Italian peninsula with islands like Sardinia, Corse, Elba, Sicily, and the south Mediterranean (Tunisia, Morocco); many traders and passengers use the quite impressive ferry network departing from Livorno.

With cruise tourism unavoidably appearing, Livorno was identified as a key port of call, a real gateway to allow tourists to briefly visit Florence, Pisa, and Lucca, important international tourism attractions., which is illustrated at Figure 4.

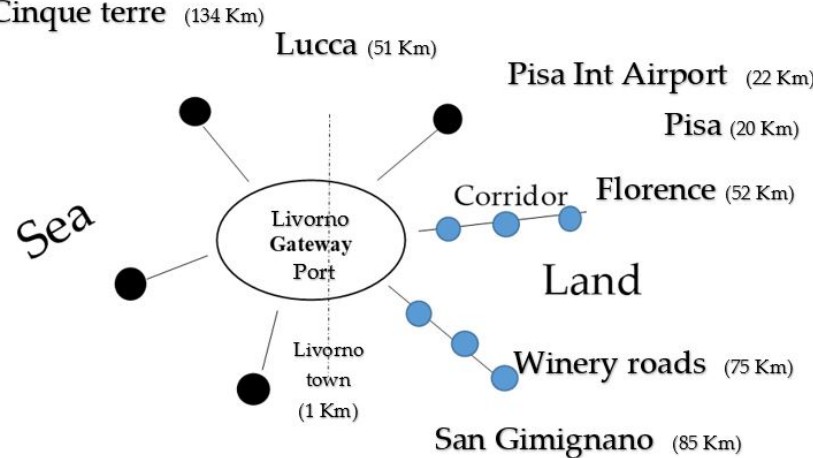

**Figure 4.** Livorno gateway port.

Livorno still has its role as a trading and ferry port. The cruise component is relatively new and accounts for 786,000 passengers, compared to the ferry movement with 2.65 million persons per year. The port has 11 berths for ferries and cruise ships, and it is basically connected with a short transit

area for passengers, with terminal features offering not just a ticket office and waiting area, but also additional services (info points, security, internet point, Wi-Fi, cafeteria, etc.).

The logistic terminal has a shuttle bus to the town center, as well as a taxi, car rental, and "fly and cruise" service. The "fly and cruise" service connects the port with Pisa airport; in principle, this great asset of being very close to an international airport could allow Livorno to take the role of a homeport. Tourists could fly into Pisa and depart for cruises without carrying their luggage. Furthermore, the role of a homeport entails many shore services for cruises, that over time were developed by other nearby ports, such as Barcelona and Genova. Among port competitors, there is already a division of roles. It is not easy to combine the role of being a traditional trading and ferry movement port without high specific investments and great stakeholder ability to displace competitors already in the homeport role.

Inland tours chosen by Livorno cruise passengers see 90% of the total excursions going to Florence, Pisa, and Lucca, while Cinque Terre accounts for 4.37% (Porto Livorno 2000, 2019). In the last few years, the inland assets offered even many other possibilities to visit the Etruscan coast and towns, including wine and oil areas, and the unique Tuscan countryside. It is a very important excursion supply that offers the opportunity to discover Tuscany's specific identity, while "decongesting" the most requested and visited towns. The latest report on tourism in Tuscany accounted Florence with more than 26 million visitors in 2018 [37].

There is a historical "path dependence" of the port of Livorno on Florence and Pisa, which seems to be confirmed by the cruise sector's choices. Now, the number of town destinations is uneven, adding people to the already crowded famous towns.

Livorno is a gateway port, and it would be profitable to try enriching this role in a conscious way in order to invest in and develop assets that can allow the town to be appealing and desirable for tourists when arriving via ferry or disembarking from cruises.

The ferry terminal and two cruise terminals in Livorno—the *Alto Fondale* and *Porto Mediceo*—which accommodate larger cruise ships and highlight the historic fort, are walking distance from downtown (only 8 min by foot). Several itineraries to explore the historical part of the city are proposed, such as for example a ride to the beautiful area with canals called *La Venezia*, built during the 15th century based on a Venetian architect's project.

Of course, to be "a first choice" in the territory, many conditions need to be developed. First of all, the proposed excursions should be promoted through a segmented marketing and communication strategy in order for Livorno to become a demanded destination. Moreover, it would be necessary to put together cruise plans with a stop lasting at least a couple of days in Livorno.

For this, we would need to discuss the cruise package decisions taken by these gigantic oligopolistic companies, which is a huge and difficult bargain for local stakeholders. It is not surprising that, despite the unusual history of Livorno, just a very small number of tourists disembark and visit the town (0.62%) [37]. The vocation of Livorno was never a tourist one and putting value into its history and its monuments, museums, and identity is a process that just recently started.

In Livorno, the local government suffers from a weak public policy and weak management practices. The integration of tourism into a local strategic planning framework is still absent; we can highlight the lack of consultation between the tourism industry, private companies, the port authority, trade organizations, and public institutions that, instead, should be essential for enhancing local economies and promoting the discovery of social and cultural context. Moreover, the new majority private property of Porto Livorno 2000 did not clarify its own strategy.

Regarding cruise sustainability means measuring and calibrating the effort and stakeholder involvement to strengthen the port's role and designing new appealing services and activities able to attract people for a longer time than the cruise stops.

## 4. Results

In light of the empirical research that led to the designing of the two case studies of Lisbon and Livorno, we developed a tentative "model" for the different ports' roles. Table 1, built from different

sources of information, such as interviews with executives and secondary data gathered by port offices and the public published by consultants and official websites, summarizes the accessibility, characteristics, and main supply of each port according to the variables presented in our theoretical framework (see Section 3).

**Table 1.** Comparison between the roles of the ports of Lisbon and Livorno.

| Variables | Livorno (Italy) | Lisbon (Portugal) |
|---|---|---|
| **Type of Port** | **Gateway Port** | **Destination Port** |
| Ownership/ management | Until 2018, the proprietary majority was public. Since 2019, 66% is privately hold by Moby (main ferry company) + 34% by the Chamber of Commerce and Port Network Authority of the Tyrrhenian Sea | The Administração do Porto de Lisboa (APL), the administration of the Lisbon Port, is an association that has the grant to the cruise terminal activity for the last 35 years. Since 2014, the APL has a private ownership. |
| Port location | -West-Med Route: Morocco–Livorno<br>-North Tyrrhenian Multi-Port Gateway<br>-Northern Italy<br>-Central–Eastern Europe<br>-West Mediterranean/Eastern Europe "land bridge"<br>-North–South America | Switching point or base port for<br>-Atlantic coast of Europe;<br>-Western Mediterranean;<br>-Northern Europe;<br>-Africa, Madeira, and the Canary Islands |
| Port core business * | Cargo shipment: 748,000 TEU movements (+1.9%)<br>Ro-Ro (cars) sector: 507,000 movements (+13.2%)<br>Energy Products: Long tradition in chemical and gas and oil sector: 11 million tons (+10%)<br>Ferry: 2.65 million passengers (+5.2%)<br>Cruises: 786,000 passengers (+12.5%) | Container cargo: 56%<br>Bulk agri-food stuffs and liquid and solid raw materials: 29%<br>Tourism and leisure: 15%<br>Cruises: 487,000 passengers (+11% compared to 2018) |
| Cruise terminal positioning | Alto Fondale and Porto Mediceo; independent management; market position as premium/luxury cruises tours.<br>In expansion, but still interstitial. | 3 cruise terminals: Alcantara, Santa Apolonia, and Jardim do Tabaco Quay.<br>Market position as a turnaround port for large international cruises companies<br>Exponential growth, but still niche business segment |
| Port infrastructures | 11 Berths for cruise ships and ferries (3.5 km of port berths)<br>2 cruise terminals<br>1 ferry terminal<br>Shore-side electric power supply plant | Port's main access channel has 14 m depth<br>1500 m of berth quay (depths between 8 m and 12 m)<br>13,800 m$^2$ of terminal facilities over 3 floors<br>1490 m of pier for multi length ships<br>3 cruise terminals (north bank of the River Tagus)<br>4 recreational docks: Alcantara, Belem, Bom Sucesso, and Santo Amaro |
| Terminal features | Waiting area<br>Info point<br>Check-in desks<br>Security check<br>Ticket offices<br>Cash dispenser<br>Internet Point<br>Wi-Fi<br>Parking<br>Bar and cafeteria<br>Shopping center<br>Self-service restaurant | Waiting area<br>Wi-Fi<br>Info tour for experiencing Lisbon<br>Duty-free stores<br>Ship storage area<br>Onsite equipment (forklift, crane, and others)<br>Supplying services (water, provisions, and others)<br>Fully automated gangway system<br>Post office<br>Public phones<br>Souvenirs shops<br>Wine shops |

**Table 1.** *Cont.*

| Variables | Livorno (Italy) | Lisbon (Portugal) |
|---|---|---|
| Type of Port | Gateway Port | Destination Port |
| Terminal logistics | Bus to Pisa international airport (fly and cruise)<br>Car rental booth (drive and cruise)<br>Taxi parking<br>Chauffeur service parking<br>Shuttle bus to Livorno city center | Connection to Lisbon railway station;<br>80 bus parking spaces at Alcantara terminal<br>Taxi parking<br>Cars rental<br>360 car parking spaces<br>Coach park<br>Shuttle bus to Lisbon city center |
| Inland assets | Florence–Pisa–Lucca (90% of total excursions)<br>Cinque Terre (4.37%)<br>San Gimignano/Volterra<br>Etruscan coast excursions, wine and oil roads, Tuscan museums, wine tasting, truffle hunting, farmhouse visiting, etc.) | City of Lisbon ** (90%)<br>Sintra (28%)<br>Cascais (17%)<br>Fatima (10%) |
| Additional services | -Event and exhibition centers<br>-Tour operator booth | -Panoramic view terrace<br>-Music concert hall |

* For Livorno, data were taken from the port authority (2019 compared with 2018); for Lisbon, the main source of information was the Lisbon port website. ** Multiple choice answers. Data released from Observatorio Turismo de Lisboa, "Survey to cruise passengers, Porto del Lisboa", 2017.

The roles that Lisbon and Livorno assume with regard to cruise packages is are as a destination port and gateway port, respectively.

Lisbon is amongst the most popular tourism destinations in the world and one of the main international cruise terminals, experiencing a growth in cruise demand [52]. People docking in Lisbon are driven by the cruise companies to visit the town. City tours are usually directly organized by the cruise line companies and promoted by their connected travel wholesalers. Instead, Livorno is considered a true "gateway" to the wonders of Tuscany (Florence, Pisa, Cinque Terre, Etruscan coast, and so on). Not surprisingly, Livorno is presented in many cruise line itineraries as the "port of Florence". The city of Livorno serves as a jump-off point for daytrips elsewhere; Florence is no doubt the primary destination, but even other cities like Pisa, Lucca, and San Gimignano are also options.

Despite the growing importance of cruise tourism and passenger traffic in Lisbon and Livorno, both towns were born as merchant ports. Lisbon has a tradition in container cargo shipment, together with the movement of solid and liquid raw materials. Nevertheless, Lisbon is globally known "city break" destination; the local attractions, the strategic location, and the mild climate stimulate a growing number of tourists to visit the town throughout the year. Thus, the tourism and leisure segments became progressively important for the port.

Livorno was historically devoted to the shipment of goods and to ferry transportation, while cruises are still an interstitial activity. Indeed, in terms of the contribution to the port town economy, in Livorno, freight traffic is the most relevant, followed by people embarking on and disembarking from ferry boats, involving three-quarters of passenger turnover, the most significant category of the Livorno port. Cruise passengers are increasing but, as of now, they account for one-quarter of total passenger traffic.

Although both harbors are in the town, in Lisbon, the different segments of port activity are very clearly separated, and the cruise ships are also in a quite isolated area from the cargo. Instead, in Livorno, where freight traffic and ferry passenger embarkment/disembarkment is still prevalent compared to cruises, this separation is not so evident, and there are some areas forbidden to pedestrians. To ease the accessibility of cruisers to the town and transport facilities (car, taxi and bus parking, car rentals, etc.), some specific investments would be required. For instance, there is an issue with cellulose raw material deposited on the trade berth too close to the passenger dock. This kind of contamination would require new security investments to protect passengers from pollution.

For both ports, cruise tourism is not their core business. In spite of that, Lisbon and Livorno largely invested in different specific proportions to extend new cruise terminals, in order to offer core and additional services. They are rebuilding and expanding the infrastructures and facilities to host larger and more numerous cruises vessels, as well as provide comfort and entertainment amenities.

In both cases, those massive investments promoting the expansion of cruise tourism seem to be strictly connected to the recent privatization of the two port authorities (PortoLivorno2000 and APL-Administração do Porto de Lisboa). Still, some differences are emerging. In Lisbon, the port is strategically investing in cruise terminals and in their positioning to appear as a unique and competitive destination, even boosting the authenticity of local businesses, such as local product providers (local shops, small restaurants, etc.). Lisbon APL is working on creating a direct networking and communication process with local associations and private cruise companies. On the contrary, in Livorno, the main interest of the port's new majority private owner is the ferry port's facilities and services in order to boost the passenger traffic in the port, instead of developing a shared wider strategy to enhance Livorno as a tourist destination. It is clear that, in both ports, the core business is freight traffic, while cruises are a growing but still niche segment of tourism.

## 5. Conclusions Remarks and Discussion

This comparative study showed similar sustainability challenges for significantly different ports. We started by analyzing the cruise sector, the ports' structure and assets, and the state of the art regarding cruise destinations, carrying capacity, and port categories.

*Socio-cultural* and *economic sustainability* occupies an important role in the cruise industry. Key concerns include the cooperation between and co-location of close ports, as well as the inhabitants' quality of life, the accessibility to recreation, the management of local infrastructures, public transportation, and road congestion, the protection of the cultural heritage of the city, the reduction of the pressure on main attraction areas through the implementation of diversified offers, the promotion of local economies (small enterprises, typical food, product manufacturing, etc.), and the control of service prices (taxis, shops, restaurants, museums, etc.).

Although cruise tourism accounts for a low percentage of the overall number of visitors, as reported by industry reports [37,50], and is, thus, a minor contributor to crowd tourism, the concept of socio-economic and cultural sustainability appears to be connected to the simultaneous presence of huge numbers of people—sometimes about 2000–4000 passengers—disembarking simultaneously and moving around in limited areas for a few hours. Furthermore, the surveys applied in the Lisbon and Livorno ports showed that cruise visitor expenditure is relatively low, which is aligned with research concerning passengers [16].

In Lisbon, the coincidence of being a mass tourism destination and a cruise port is especially delicate in terms of sustainability. The city government is facing an impressive increase in tourism flow, even higher than Amsterdam or Barcelona. In response to this situation, Lisbon is undertaking a leading role in the process of coordination and exchange with multiple actors involved in the tourism *filiére* (private companies, cruise associations, port authority, local government, not-for-profit organizations, etc.), trying to intervene in the regional and local tourism planning and mass tourism management [52].

Conversely, the city of *Livorno* is an *intermediate location*; it developed a unique geography where its importance is derived from its accessibility (as a gateway to inland and/or air transportation through the Pisa and Florence airports) rather than its town's intrinsic characteristics. A port like Livorno has advantages deriving from its logistic strategic position, as well as providing access to more attractive surrounding places. It is a port-city which is not yet considered a well-known tourist attraction, and it is identified as "very interesting for its genuineness". In preserving that, Livorno would need to support its competitiveness through a balance of local and international tourist operators. This could help reduce the onshore excursions to crowded sites (e.g., Florence), diversifying the cruise excursion destinations. It would be strategic to include the port's historic town and other Tuscan attractions in a customized way, building a supply able to attract more selective and demanding visitors. Livorno

should offer a distinctive "cruise shore scape", i.e., integrated land-based components of both the urban port and adjacent hinterland [19], in order to propose an authentic inland experience.

The strategic vision, the revitalization of the port infrastructure, and the innovative communication campaigns by cruise companies and industry associations and travel global operators, undertaken by port terminal Livorno 2000 in order to attract cruise tourists to its historic parts, resulted in increased local visits and longer cruise ships stays. In Lisbon port, the active participation in trade international meetings is also allowing cruise tourism development. However, the government actions of these two ports reflect a cooperation among stakeholders, in alignment with strategies considered to change the ports' configurations and achieve an upper positioning as hub ports [44,46]. Also, from the sustainability point of view, the research builds on previous research, highlighting that the identification of the port role in the cruise *filiére* is a key aspect to understanding where the high numbers of people disembarking are spending their time inland; however, this research also contributed through a comparative approach of two quite different ports in their characteristics but with quite similar strategies and management actions (attracting cruise ships in a responsible way, as well as reducing crowd visitors to be most demanded touristic places).

From both cases we analyzed, it emerges that the most demanded tourist towns (Lisbon and Florence) suffer from "*over-tourism*" with a carrying capacity which is close to collapse regarding services for visitors and the quality of life of the local citizens. A possible strategy could be, therefore, the implementation of a kind of "coopetition" with other local places, in order to try spread the number of visitors among inland cities. Respecting the specificities of the cruise port towns, creating a joint identity with its surroundings, and proposing dynamic experiences and routes for niche visitors should be effective marketing strategies for Lisbon and Livorno/Tuscany in order to find a balance between challenges, such as visitor pressure, and caring both for the local community and the destination's stakeholders.

Furthermore, we suggest that cruise visitors and local brands could share and experience, in the terminal, a quickly available sample of the best regional experiences in gastronomy, products, and culture. This could motivate longer and repeated tourist visits, while pleasantly enjoying the regional offers. We suggest that a much deeper cruise terminal concept designed as a top-quality and genuine sample of the products and gastronomy from the region would add value to both residents and visitors. Indeed, a new cruise terminal concept should combine all the above, reducing the percentage of land services for cruise lines [41] and increasing the safety of cruise passengers. This emerging trend of combining port terminals, local offers, and city congestion was suggested in some previous studies [19,42] and from the innovation in some ports [8] or the building of new ones [7], where they adopt the duty-free style from airport shopping.

## 6. Study Limitations and Future Research

The main limitation stems from the fact that the questionnaires were not designed to answer the questions raised by the cruise tourism literature, and our focus was instead on the ports' role in achieving greater sustainability. We hope that this study and methodology inspires other researchers in designing more specific databases and extending the study to other locations and experiments.

Specific data on cruise tourism segmentation and consumption were not available. In both ports, the surveys prepared by the authorities show the passengers' socio-demographic and nationality profiles, as well as their cruise trip choice motivations and behavior aligned with industry reports such as CLIA and Med Cruise, in addition to previous research focusing on visitors. The majority of passengers experienced cruise tourism, booked the trip mainly from a travel agency to the Mediterranean or Caribbean lines, staying about two weeks on the cruise trip, where the itinerary was a key motivation, and in which cultural visits and shopping accounted for the major consumption of time and cost when inland, showing an overall high level of satisfaction, although it was lower in the terminal services and offices in Livorno.

However, we could not compare and carry out a cross-analysis between the segments' expectations, or evaluations of the ports' services (entertainment and shopping) and the towns' offers.

A quantitative study on the importance of reducing crowds and the number of visits to the town's port services and infrastructures was obtained from a survey addressed to the passengers, as well as the industry stakeholders (cruise companies and tourism operators). Although, in both ports' terminals, managers are attracting cruise ships to increase the length of stay, on average, cruise visitors do not stay long enough to find the characteristics and genuine products and brands of the town. These findings were in conformity with academic research [22]. Also, these studies quantified the cost/benefits for each stakeholder (cruise companies, businesses, and ports terminal management), or at least to have a deeper perspective of the other two stakeholders regarding the potential use and concept designed for the cruise terminal's commercial and entertainment area. Studies on how to benefit from the port areas for local businesses (public and private ones) and port terminals is a developing research stream [26]. It would be important to analyze how the ports' cruise terminals could be a tool to reduce congestion for shorter visits.

Furthermore, an understanding of the passengers' and residents' opinions concerning the value added in the ports' role with regard to the co-destination concept between cruise ships and ports of call, as introduced by Whyte et al. (2018) [42], can enhance the knowledge on port policy that can reach a sustainable future for the industry.

**Author Contributions:** This article is the result of joint work by all authors, who equally contributed to the design and research. All authors collaborated in analyzing data, preparing the data, and writing the paper. All authors discussed and agreed to submit the manuscript. Conceptualization, M.S., E.R. and P.Z.; data curation, M.S., E.R. and P.Z.; resources, M.S., E.R. and P.Z.; writing—original draft, M.S.; writing—review and editing, E.R. and P.Z.

**Funding:** This research was supported by Fundação para a Ciência e a Tecnologia, grant UID/GES/00315/2019.

**Acknowledgments:** This study would not be possible without the valuable support from the personal face-to-face interviews with the managers of the Port of Lisbon and the Port of Livorno, as well as industry experts.

**Conflicts of Interest:** The authors declare no conflicts of interest.

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
