# Peer review of "Port’s Role as a Determinant of Cruise Destination Socio-Economic Sustainability"

_sustainability, doi:10.3390/su11174542_

Round 1

Reviewer 1 Report

I found the originality of the article to be significant. Not only is the reader introduced to a new place, but a new (to me) method for understanding various metrics for sustainability and quantification and valorization of various practices. The presentation of the material is mostly appropriate, though I suggest below numerous places where it can be improved. As much as anything, I found the information to be presented in a fashion that makes it both understandable and interesting to the journal readership. My overall recommendation is that the manuscript be accepted with minor revisions. The manuscript is fundamentally a positive contribution to the literature, though I suggest some changes to improve the impact of the research.

The literature review needs to be better integrated into the paper. What is the purpose of the literature review? How are the topics covered in the literature review justified as relevant? I suggest adding a statement of purpose for the literature review as well as an outline of themes/topics with a justification for each. The literature review would also benefit from subheadings.

The discussion and conclusions do not provide enough detail. Also, the discussion section could benefit from an engagement with the literature that addresses the novelty of the approach, the significance of integrating models and how it might contribute to broader knowledge in the field.

While I suggest these relatively minor revisions of the article, I remain excited about the manuscript’s contribution to Sustainability. I hope the authors receive this review and strengthen the manuscript, and I look forward to seeing it in publication.

Author Response

Dear editors,

We are very grateful for the many valuable comments and suggestions that three anonymous referees and you have provided to our paper. We agree with most of the suggestions and tried to incorporate them, as much as possible, in the version that we are resubmitting.

Responding to a still understudied influence of the port and its relation with the city and the global cruises and travel operators (see for example Cussano, et al, 2018, reference 30), the aim of our study is to contribute to the solution of one of the most critical challenges to the cruise industry, aggravated by the rising size of cruise ships – port and near cities congestion and lack of integration with the local economy and culture. Our study innovatesfocusing on the actions that the port terminal management can help on that, includingthe ports’ managersperspective, as well as observingthe ports’ managersactions in order to integrate local businessesin the terminal cruisefacilitiesandin the onshoretours range.This study contributes toanalyzehow port managersmaybridge the relations betweenlocal businesses and global companies, as well as attract land-visitors to the terminal cruises facilities andsohelp reducingtourismcongestion.

Furthermore, our article contributes to cross the following literature branches: (1) the integration port-city (for example Weaver & Lawton, 2017 , the reference 21 or McCarthy, 2018, the reference 7); (2) cooperation among stakeholders, namely the cruise companies (see for example London & Lohman, 2014, Bags& Doomis, 2014)  and (3) the socio-economic cost/benefits trade-off (see for example MacNeill & Wozniak, 2018- ref 11 - or Klein, 2011- ref. 28). But analyzing under the context of the crowd tourism perception trend, which recently was studied on the cruise tourism onshore in the Sustainability Journal (Sanz-Blas et al, 2019- ref. 15).

Finally, although the methodologic approach, combining the port managers ‘depth interviews with ports observation and secondary quantitative industry studies, do give our study an exploratory approach, but also shows the actions undertaken by these two ports, which complement previous research, such as the study of Marques & Cruse, 2015 (ref. 42).

For this propose we considered two ports located in the same “Mediterranean Route”, but substantially different in key dimensions. Livorno is a traditional port that serves mostly as a gateway to key tourist attractions in extremely reputed towns such as Florence. Complex logistics are required to carry thousands of passengers who then briefly flood coffee houses, restaurants, stores museums, etc., in competition with many tourists arriving by other means. By contrast, Lisbon is relatively new to the cruise industry and, indeed, to mass tourism, as other Portuguese locations such as Algarve and Madeira used to attract the bulk of foreign and even domestic tourists. Strong efforts to diversify the tourism offer, beyond sun and sea, have gradually succeeded in attracting tourists to other locations, both along the coast and inland. This was done through better promotion and the development of infrastructure, while many private sector hotels have been built to meet the rising demand. Cruise tourism was attracted through a modern infrastructure located near the historical center.

Our study was started with a literature review, including many empirical studies on the Caribbean cruises, the number one cruise region. This was followed by in depth structured interviews to the managers of both ports and the analysis of questionnaires to large samples of cruise customers.

After receiving your suggestions, we went through a systematic revision of the text, including both content and format, as you can find in the “track changes” version attached. We are also including a “clean” version that we hope will merit your acceptance for “Sustainability”.

We tried to “strengthen the methodology and clarify the research aims”. The abstract, in particular, was thoroughly revised, hoping to make it more objective and clearer. The whole text was revised and, in several parts, rewritten, to improve the “links among research aims-analysis-results”. The conclusion includes now a deeper discussion and future research suggestions, aligned with literature state of art. It tries to capture the main results and potential implications for both public authorities and the business community, while leaving suggestions for future research.

We will now briefly address the comments and suggestions provided by the referees.

Referee 1

While providing encouraging feed-back and a positive assessment of the originality and merit of the study, the referee makes suggestions for minor revisions, better integration and structuring of the literature and the need for more detailed discussion and conclusions. We hope referee 1 can find that we have positively met his expectations, as described before.

Reviewer 2 Report

Thank you very much for the opportunity to review this work, which is interesting. I believe that research on cruise tourism is important, since the economic, social, and environmental impacts of this activity must be studied. Impacts that can be positive or negative.

It would be good if the authors of this work improve some aspects of it.

- I believe that the objective of the investigation should be exposed in a simpler and clearer way in the introduction. It should be noted what was not known about this topic, and what will be known once this work is done.

- Likewise, the methodology should be stated more clearly in the introduction, it should be specified which analysis techniques will be used in a concrete way and why they have been selected.

- I think the title is not completely representative of the objective of the work.

- In the methodology section, the review of the literature is exposed, which should not be part of a methodology section.

- In the literature review should indicate the research gap on which you are working. Are there previous studies on this field?

- The description of the methodology is very scarce, it is not well justified, the data processing process is not detailed.

- I do not find a methodology applied correctly. In the results section only a description of the ports appears.

- With the above, I don´t understand what you intend to get with this article or how.

- There is no clear sequence of objectives, research gap, methodology, application and results.

Please describe a goal in detail. Explain what methodology you will use to achieve the objective. Justify the methodology and explain the data you will use. Indicates what the research gap is. It details the way in which the methodology has been applied to the data to achieve the objective.

This work needs to be reviewed in full in order to be published.

Kind regards

Author Response

Dear editors,

We are very grateful for the many valuable comments and suggestions that three anonymous referees and you have provided to our paper. We agree with most of the suggestions and tried to incorporate them, as much as possible, in the version that we are resubmitting.

Responding to a still understudied influence of the port and its relation with the city and the global cruises and travel operators (see for example Cussano, et al, 2018, reference 30), the aim of our study is to contribute to the solution of one of the most critical challenges to the cruise industry, aggravated by the rising size of cruise ships – port and near cities congestion and lack of integration with the local economy and culture. Our study innovatesfocusing on the actions that the port terminal management can help on that, includingthe ports’ managersperspective, as well as observingthe ports’ managersactions in order to integrate local businessesin the terminal cruisefacilitiesandin the onshoretours range.This study contributes toanalyzehow port managersmaybridge the relations betweenlocal businesses and global companies, as well as attract land-visitors to the terminal cruises facilities andsohelp reducingtourismcongestion.

Furthermore, our article contributes to cross the following literature branches: (1) the integration port-city (for example Weaver & Lawton, 2017 , the reference 21 or McCarthy, 2018, the reference 7); (2) cooperation among stakeholders, namely the cruise companies (see for example London & Lohman, 2014, Bags& Doomis, 2014)  and (3) the socio-economic cost/benefits trade-off (see for example MacNeill & Wozniak, 2018- ref 11 - or Klein, 2011- ref. 28). But analyzing under the context of the crowd tourism perception trend, which recently was studied on the cruise tourism onshore in the Sustainability Journal (Sanz-Blas et al, 2019- ref. 15).

Finally, although the methodologic approach, combining the port managers ‘depth interviews with ports observation and secondary quantitative industry studies, do give our study an exploratory approach, but also shows the actions undertaken by these two ports, which complement previous research, such as the study of Marques & Cruse, 2015 (ref. 42).

For this propose we considered two ports located in the same “Mediterranean Route”, but substantially different in key dimensions. Livorno is a traditional port that serves mostly as a gateway to key tourist attractions in extremely reputed towns such as Florence. Complex logistics are required to carry thousands of passengers who then briefly flood coffee houses, restaurants, stores museums, etc., in competition with many tourists arriving by other means. By contrast, Lisbon is relatively new to the cruise industry and, indeed, to mass tourism, as other Portuguese locations such as Algarve and Madeira used to attract the bulk of foreign and even domestic tourists. Strong efforts to diversify the tourism offer, beyond sun and sea, have gradually succeeded in attracting tourists to other locations, both along the coast and inland. This was done through better promotion and the development of infrastructure, while many private sector hotels have been built to meet the rising demand. Cruise tourism was attracted through a modern infrastructure located near the historical center.

Our study was started with a literature review, including many empirical studies on the Caribbean cruises, the number one cruise region. This was followed by in depth structured interviews to the managers of both ports and the analysis of questionnaires to large samples of cruise customers.

After receiving your suggestions, we went through a systematic revision of the text, including both content and format, as you can find in the “track changes” version attached. We are also including a “clean” version that we hope will merit your acceptance for “Sustainability”.

We tried to “strengthen the methodology and clarify the research aims”. The abstract, in particular, was thoroughly revised, hoping to make it more objective and clearer. The whole text was revised and, in several parts, rewritten, to improve the “links among research aims-analysis-results”. The conclusion includes now a deeper discussion and future research suggestions, aligned with literature state of art. It tries to capture the main results and potential implications for both public authorities and the business community, while leaving suggestions for future research.

We will now briefly address the comments and suggestions provided by the referees.

Referee 2 also makes a positive assessment of the relevance and opportunity of the topic, although requesting more substantial improvements. He/she starts by emphasizing the need for a clearer exposure of the objective and the methodology of the investigation, also a request like the referee 1’s. Referee 2 also suggests:

-        Finding a more representative title. We changed the title.

-        We tried to have more clear boundaries between literature review and methodology.

-        We hope the research gap is better identified. Although the sustainability and congestion challenges to cruises has been analyzed in the literature, the different profile of ports has been neglected;

-        We tried to make the methodology and identification of key variables more explicit;

-        Finally, we tried to make a clearer conclusion, as well as relevant recommendations regarding the congestion avoidance, engagement of the local business community overall increase in the satisfaction of the cruise tourists and opportunities for future research with a more specific data collection that allows the finding of specific demands for stopping length and visiting preferences while docked. Cruise tourists with different patterns (age, gender, income, etc.) may have different preferences, an information that can be crucial for cruise operators marketing and segmentation.

We also hope referee 2 finds the overall revision of the study adequate.

Reviewer 3 Report

This is a descriptive paper, based on a rather generic analysis and discussion of a couple of ports.  The authors have failed to adequately review the relevant recent literature on cruise tourism, which is refelcted in the generic conclusions that they arrive to.  

For example:

Line 591:  "At the end of this comparative study, emerge that the key aspect of sustainability in Tourist destination is quite impressive."

Line 597:  "The concept of sustainability seems to occupy an important role in cruise Industry"

By simply reading through a couple of cruise papers in tourism journals, you would have quickly reached the same conclusion!

This is a rather simplistic and (frankly) not particularly useful conclusion.  If the authors would have familiarised themselves just slightly with cruise tourism literature and some of the key Researchers in the area, they could have perhaps elaborated and extracted something useful from the cases.  The case-study analysis itself follows no clear methodological (qualitative) paradigm and the value of using a comparative approach is neither clear nor pertinent (in terms of scope - only 2 cases which are different).  Finally, the text needs some major language editing.  

In all aspects - literature review, research methodology, discussion and conclusions - this paper is scientifically sub-standard, simply re-interating 'basic' cruise tourism knowledge and providing generic statements as conlcusions   

Author Response

Dear editors,

We are very grateful for the many valuable comments and suggestions that three anonymous referees and you have provided to our paper. We agree with most of the suggestions and tried to incorporate them, as much as possible, in the version that we are resubmitting.

Responding to a still understudied influence of the port and its relation with the city and the global cruises and travel operators (see for example Cussano, et al, 2018, reference 30), the aim of our study is to contribute to the solution of one of the most critical challenges to the cruise industry, aggravated by the rising size of cruise ships – port and near cities congestion and lack of integration with the local economy and culture. Our study innovatesfocusing on the actions that the port terminal management can help on that, includingthe ports’ managersperspective, as well as observingthe ports’ managersactions in order to integrate local businessesin the terminal cruisefacilitiesandin the onshoretours range.This study contributes toanalyzehow port managersmaybridge the relations betweenlocal businesses and global companies, as well as attract land-visitors to the terminal cruises facilities andsohelp reducingtourismcongestion.

Furthermore, our article contributes to cross the following literature branches: (1) the integration port-city (for example Weaver & Lawton, 2017 , the reference 21 or McCarthy, 2018, the reference 7); (2) cooperation among stakeholders, namely the cruise companies (see for example London & Lohman, 2014, Bags& Doomis, 2014)  and (3) the socio-economic cost/benefits trade-off (see for example MacNeill & Wozniak, 2018- ref 11 - or Klein, 2011- ref. 28). But analyzing under the context of the crowd tourism perception trend, which recently was studied on the cruise tourism onshore in the Sustainability Journal (Sanz-Blas et al, 2019- ref. 15).

Finally, although the methodologic approach, combining the port managers ‘depth interviews with ports observation and secondary quantitative industry studies, do give our study an exploratory approach, but also shows the actions undertaken by these two ports, which complement previous research, such as the study of Marques & Cruse, 2015 (ref. 42).

For this propose we considered two ports located in the same “Mediterranean Route”, but substantially different in key dimensions. Livorno is a traditional port that serves mostly as a gateway to key tourist attractions in extremely reputed towns such as Florence. Complex logistics are required to carry thousands of passengers who then briefly flood coffee houses, restaurants, stores museums, etc., in competition with many tourists arriving by other means. By contrast, Lisbon is relatively new to the cruise industry and, indeed, to mass tourism, as other Portuguese locations such as Algarve and Madeira used to attract the bulk of foreign and even domestic tourists. Strong efforts to diversify the tourism offer, beyond sun and sea, have gradually succeeded in attracting tourists to other locations, both along the coast and inland. This was done through better promotion and the development of infrastructure, while many private sector hotels have been built to meet the rising demand. Cruise tourism was attracted through a modern infrastructure located near the historical center.

Our study was started with a literature review, including many empirical studies on the Caribbean cruises, the number one cruise region. This was followed by in depth structured interviews to the managers of both ports and the analysis of questionnaires to large samples of cruise customers.

After receiving your suggestions, we went through a systematic revision of the text, including both content and format, as you can find in the “track changes” version attached. We are also including a “clean” version that we hope will merit your acceptance for “Sustainability”.

We tried to “strengthen the methodology and clarify the research aims”. The abstract, in particular, was thoroughly revised, hoping to make it more objective and clearer. The whole text was revised and, in several parts, rewritten, to improve the “links among research aims-analysis-results”. The conclusion includes now a deeper discussion and future research suggestions, aligned with literature state of art. It tries to capture the main results and potential implications for both public authorities and the business community, while leaving suggestions for future research.

Referee 3 is critical of the literature review considering that relevant literature was ignored in this study. We have so included several references that had not been included in the earlier version of the study.

We entirely agree with the criticism to the previous sentences in lines 591 and 597. We were meaning that sustainability is a key concern for the cruise industry, as regulators, public authorities, residents and the tourists are aware of the challenges faced by this industry. We did not mean, however that that fully appropriate mitigating measures had already been undertaken. Indeed, the objective of this study is precisely to underline the potential steps that can be made according to the ports’ specificity.

We hope that referee 3 can agree that the choice of these two ports, that required visits to the two ports administrations and the analysis of large questionnaires and other secondary data, and a joint effort, cultural and linguistic, by the three authors, can shed some light on the policies that can improve the sustainability of an otherwise very promising industry.     

Round 2

Reviewer 2 Report

The authors have made a great effort to improve the article. I believe that the article can be published in its present version.

Reviewer 3 Report

The authors improved on the previously submitted version by elaborating their arguments and incorporating additional literature.  Yet, the main issues limiting this paper remain.  For one, the scope and results of the case-study analysis do not address or reveal any novel aspects; and thus do not justify and warrant a qualitative approach.  Secondly, a qualitative approach (case-study) should at least include a complete list of sources, plus evidence of a coding process.  Here, the qualitative analysis is outlined in rather vague and generic terms.